# Neuron-glia signaling in developing retina mediated by neurotransmitter spillover

Juliana M Rosa[1†], Rémi Bos[1†], Georgeann S Sack[1], Cécile Fortuny[2], Amit Agarwal[3], Dwight E Bergles[3†], John G Flannery[1,2†], Marla B Feller[1,4*]

[1]Department of Molecular and Cell Biology, University of California, Berkeley, Berkeley, United States; [2]Vision Science Graduate Program, University of California, Berkeley, Berkeley, United States; [3]Solomon H. Snyder Department of Neuroscience, Johns Hopkins School of Medicine, Baltimore, United States; [4]Helen Wills Neuroscience Institute, University of California, Berkeley, Berkeley, United States

**Abstract** Neuron-glia interactions play a critical role in the maturation of neural circuits; however, little is known about the pathways that mediate their communication in the developing CNS. We investigated neuron-glia signaling in the developing retina, where we demonstrate that retinal waves reliably induce calcium transients in Müller glial cells (MCs). During cholinergic waves, MC calcium transients were blocked by muscarinic acetylcholine receptor antagonists, whereas during glutamatergic waves, MC calcium transients were inhibited by ionotropic glutamate receptor antagonists, indicating that the responsiveness of MCs changes to match the neurotransmitter used to support retinal waves. Using an optical glutamate sensor we show that the decline in MC calcium transients is caused by a reduction in the amount of glutamate reaching MCs. Together, these studies indicate that neurons and MCs exhibit correlated activity during a critical period of retinal maturation that is enabled by neurotransmitter spillover from retinal synapses.

*For correspondence: mfeller@berkeley.edu

†These authors contributed equally to this work

Competing interests: The authors declare that no competing interests exist.

## Introduction

There is rich research history demonstrating the important role glial cells play in developing and maintaining neural circuits (*Garrido, 2013*). In the mature brain, glial cells provide structural scaffolding and create physical barriers that compartmentalize synapses and limit diffusion of neurotransmitters. In addition, glial cells express a variety of neurotransmitter receptors. Activation of glial receptors causes increases in intracellular calcium that in turn may lead to the secretion of factors that influence circuit function (*Haydon and Nedergaard, 2015*). In the developing brain, glial cells secrete factors that are important for neuronal differentiation, migration, axonal pathfinding and synaptogenesis (*Clarke and Barres, 2013*; *Garrido, 2013*; *Chung et al., 2015*). However, the mechanisms that are used for neuron-glial signaling in the developing CNS remain poorly defined.

There are three types of glial cells in the retina: Müller cells (MCs), astrocytes, and microglial cells. MCs, the principal glial cells of the retina, stretch radially across its thickness from the outer limiting membrane to the inner limiting membrane, sending processes throughout the inner and outer plexiform layers. MC processes envelop synapses of every neuron type, placing these cells in an ideal position to receive and respond to neuronal signals (*Newman and Reichenbach, 1996*; *Newman, 2004*, *2005*; *Reichenbach and Bringmann, 2013*). In adult retina, MCs maintain retinal homeostasis and integrity (*Byrne et al., 2013*), and more recently they have been implicated in delivery of retinol to replenish cone opsin chromophore (*Xue et al., 2015*). In zebrafish, MC ablation does not affect the laminar organization of the inner plexiform layer (IPL, *Randlett et al., 2013*) or the structure or organization of synapses in the outer plexiform layer (*Williams et al., 2010*), in sharp contrast to ablation of MCs in developing (*Dubois-Dauphin et al., 2000*) and adult mice (*Byrne et al., 2013*; *Shen et al., 2012*), which

**eLife digest** A structure at the back of the eye known as the retina is essential for vision. When light hits the retina, cells called neurons produce electrical signals that lead to the release of chemicals known as neurotransmitters. When these chemicals reach a neighboring neuron, a corresponding electrical signal is produced in this cell. In this way, information about what we can see is ultimately transmitted to the brain.

The retina also contains cells that are not neurons, such as Müller glial cells. These cells span the thickness of the retina, and appear well placed to interact with all of the different types of neuron in the retina. Other types of glial cells help the neural circuits in the brain to develop, but it is not clear whether Müller cells perform the same role in the developing retina.

Newborn mice do not open their eyes until around two weeks after they are born, during which time their retina continues to develop. The retinal neurons are already active throughout this period, and spontaneously generate periodic bursts of electrical activity that spreads across the developing retina in waves. Now, Rosa, Bos et al. show that in newborn mice, these waves also trigger brief, or 'transient', increases in the concentration of calcium ions inside Müller cells. Exactly how the Müller cells respond depends on which neurotransmitters are released during waves.

A few days before the mice open their eyes, the number of calcium transients in Müller cells decreases sharply. At the same time, neurons continue to spontaneously release waves of one particular neurotransmitter called glutamate. Rosa, Bos et al. used a sensor that showed where glutamate is found in the developing retina. This revealed that the decrease in calcium transients as the retina matures is due to less glutamate reaching the Müller cells.

These findings reveal that Müller cells are involved in the spontaneous electrical activity seen in the retina during a critical period of retinal development. The next challenge is to determine how this neuronal-glial cell signaling helps the retina to mature.

leads to dramatic rearrangements retinal structure (**Byrne et al., 2013**). However, unlike radial glial cells in the cortex and cerebellum, the role of MCs in retinal development has not been fully explored.

MCs are the last retinal cell types to mature, integrating into neuronal circuits during the first two postnatal weeks (**Jadhav et al., 2009**; **Williams et al., 2010**) when the inner retina is undergoing robust synaptogenesis. During this developmental period, the retina is spontaneously active, and ganglion cells (the output cells of the retina) exhibit depolarizations that spread laterally across the retina, a phenomenon termed retinal waves. These waves are mediated by circuits that change as the retina matures (**Blankenship and Feller, 2010**). From postnatal day 1 (P1) to P9, retinal waves are mediated by cholinergic circuits. As bipolar cells integrate into the circuit (P10–P14), the waves become mediated by glutamatergic signaling. At eye opening (P14), light activated signaling begins and retinal waves cease. Both cholinergic (**Ford et al., 2012**) and glutamatergic (**Blankenship and Feller, 2010**; **Firl et al., 2013**) retinal waves are associated with volume release of neurotransmitter, where the neurotransmitter is not restricted to the synaptic cleft, but spills out into the extracellular space, allowing activation of receptors on nearby ganglion cells. Hence, these waves provide a robust source of neurotransmitter that could mediate periodic activation of MCs during retinal development.

Here we investigated if MCs participate in retinal waves as they integrate into neuronal circuits. We simultaneously performed whole-cell recordings of retinal ganglion cells (RGCs) and two-photon imaging of calcium indicators or glutamate sensors expressed specifically in MCs in order to investigate if neurotransmitters released during retinal waves reliably induce calcium transients in MCs. In addition, we determined which receptors mediate these neuron-glia interactions. Our results indicate that MCs express a complement of receptors that allow detection of neurotransmitters released during retinal waves, resulting in parallel activation of this glial network prior to eye opening.

## Results

### Developing MCs exhibit calcium transients that correlate with retinal waves

MCs undergo a dramatic morphological transformation during the second postnatal week, a time when there are robust retinal waves. At P7 (**Figure 1B**, bottom), a MC displays a simple morphology

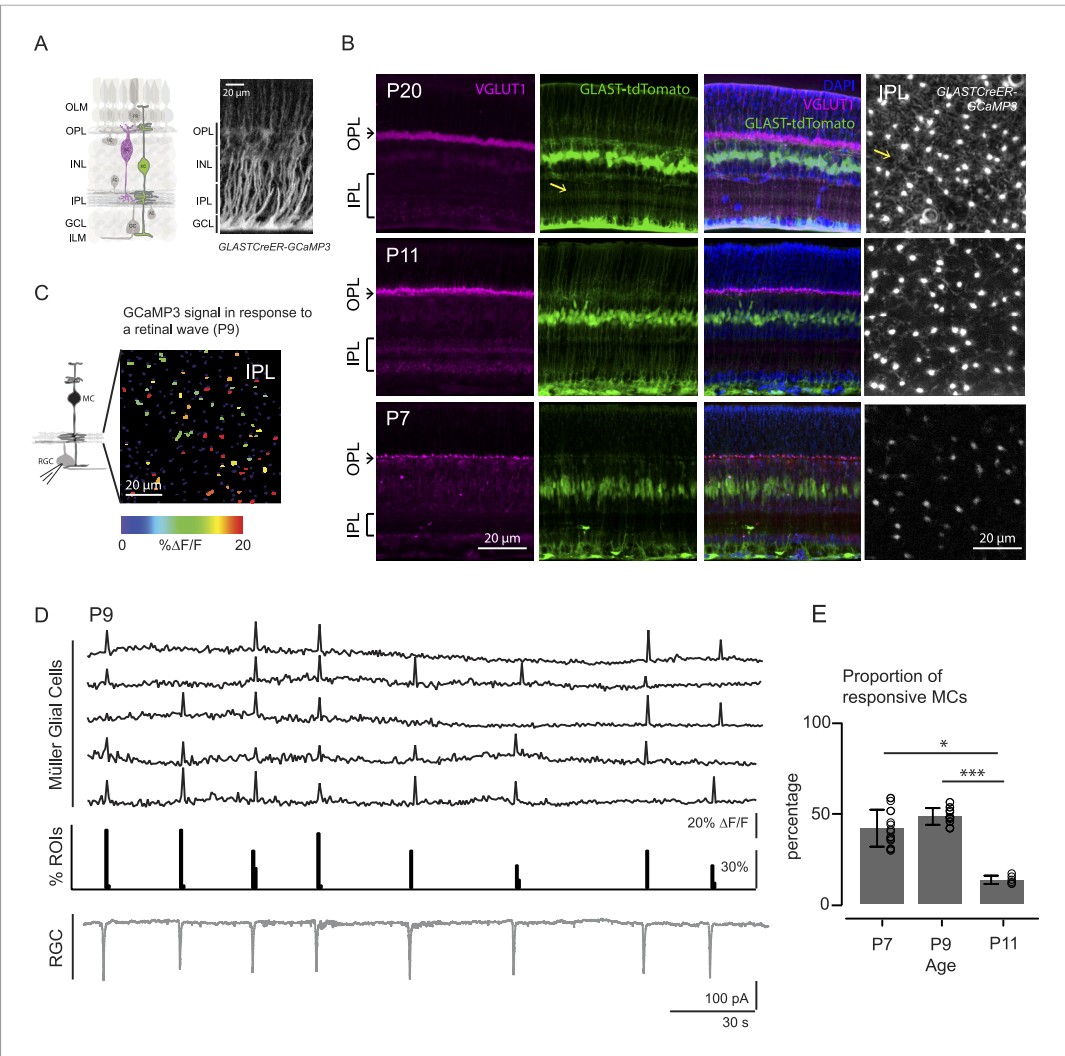

**Figure 1**. Morphology of Müller glial cells and their interactions with neurons change over development. (**A**) *Left*, Diagram of adult retinal cross-section illustrates layered circuitry (OLM: outer limiting membrane; OPL: outer plexiform layer; INL: inner nuclear layer; IPL: inner plexiform layer; GCL: ganglion cell layer; ILM: inner limiting membrane) and main cell types (PR: photoreceptor; HC: horizontal cell; BC: bipolar cell; MC: Müller glial cell; AC: amacrine cell; GC: ganglion cell). *Right*, Orthogonal projection of two-photon Z-stacks shows GCaMP3 expression in MCs of a P22 GLAST*CreER::GCaMP3* mouse retina. (**B**) Vibratome sections of GLAST*CreER::tdTomato* retinas show the structure of MCs (green; tdTomato) and the expression of vesicular glutamate transporter 1 (VGLUT1) in bipolar cells (magenta; anti-VGLUT1) at different ages. Blue stain is 4',6-diamidino-2-phenylindoele (DAPI) for visualizing cell nuclei as landmarks. Rightmost images are XY planes of the IPLs in GLAST*CreER::GCaMP3* retinas showing GCaMP3 signal at different ages. Note the expansion of MC lateral processes into the IPL with development. Yellow arrows indicate lateral processes of the Müller glial cells. (**C**) *Left*, Circuit diagram of the retina highlights cells recorded for figures **C** and **D**; labeling as in *Figure 1A*. *Right*, XY plane shows GCaMP3 signals of MCs in response to a retinal wave in a P9 GLAST*CreER::GCaMP3* retina. Color scale indicates normalized changes in fluorescence during a retinal wave. (**D**) Simultaneous MC calcium imaging and retinal ganglion cell (RGC) whole-cell voltage-clamp recording ($V_m = -60$ mV) of a P9 GLAST*CreER::GCaMP3* retina. Sample ΔF/F traces (black traces) from individual regions of interests (ROIs) (that include stalks and processes of the MC population) in response to neuronal waves recorded in a RGC (grey trace). Histogram in middle denotes percentage over time of ROIs with responsive MCs. (**E**) Percentage of ROIs with responsive MCs during at least one retinal wave at different ages. P7: 1326 ROIs from 11 retinas; P9: 3027 ROIs from 14 retinas; P11: 872 ROIs from 6 retinas. Kruskal–Wallis one-way ANOVA, Dunn's *post-hoc* test. ***p < 0.001 and *p < 0.05. See also *Figure 1—figure supplement 1* and *Video 1*.

The following figure supplement is available for figure 1:

**Figure supplement 1**. Comparison of two-photon calcium imaging signals in stalks and lateral processes of MCs.

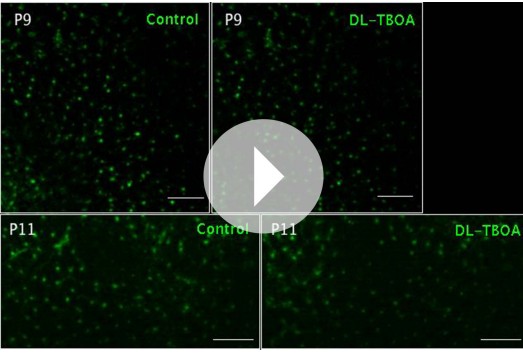

**Video 1.** Wave-induced responses are shown as changes in fluorescence of the calcium indicator GCaMP3 expressed specifically in MCs in a P9 or P11 mouse retina in the presence of the glutamate uptake blocker DL-TBOA (25 µM). Electrophysiological recordings confirmed that calcium signals were correlated with RGC activity during retinal waves. Scale bars are 20 µm. Related to *Figure 1*.

with its cell body residing in the inner nuclear layer (INL) and a thin, fiber-like process traversing the retina to form part of the inner and outer limiting membranes (green signal in GLAST:: tdTomato, middle panel, and bright punctas in GLAST*CreER::GCaMP3*, right panel). By P11 (*Figure 1B*, middle), MCs have extended many lateral processes (yellow arrows) into the IPL and these continue to grow into the adult form where they closely appose synapses and participate in removing neurotransmitters from the extracellular space (*Pow and Crook, 1996*).

Like other types of glial cells, MCs in the mature retina exhibit calcium transients that occur spontaneously or are evoked by various stimuli (*Metea and Newman, 2006*; *Reichenbach and Bringmann, 2013*). However, MC calcium signaling during retinal development remains unexplored. Thus, we first investigated if MC calcium transients during early development coincided with retinal waves. We visualized MC transients via two-photon calcium imaging of whole-mount retinas from mice expressing a genetically encoded calcium indicator specifically in MCs (GLAST-*CreER::ROSA26-lsl-GCaMP3* [GLAST-*CreER::GCaMP3*]; *Figure 1A*; see 'Materials and methods', [*Paukert et al., 2014*]). In the retina of GLAST-*CreER* mice, tamoxifen inducible Cre recombinase (CreER) is expressed by MCs (*Figure 1A*). GLAST-*CreER::GCaMP3* mice expressed sufficient GCaMP3 to detect MC calcium transients in the IPL as early as P7. Retinal waves were identified by the occurrence of compound postsynaptic excitatory currents in RGCs. Simultaneous two-photon imaging of MCs and whole-cell recordings from RGCs (*Figure 1C,D*) showed periodic MC calcium transients in the stalks and lateral processes in the IPL (*Figure 1—figure supplement 1*, *Video 1*) that coincided with RGC compound postsynaptic excitatory currents (*Blankenship et al., 2009*). Since the stalks and processes of MCs exhibited similar calcium responses, we pooled their results together throughout this study. We detected no wave-evoked calcium transients in other parts of MCs outside the IPL (i.e., in their somata, data not shown). The percentage of regions of interest (ROIs, which correspond to compartments of individual MCs, see *Figure 1—figure supplement 1*) that responded to a wave (termed responsive MCs) was high at P7 (42 ± 10.2%, 1326 ROIs from 11 retinas) and at P9 (48 ± 4.6%, 3027 ROIs from 14 retinas), but significantly lower at P11 (13 ± 2.2%, 872 ROIs from 6 retinas, *Figure 1E*). As MCs express a variety of neurotransmitter receptors, including glutamatergic and cholinergic receptors (*Wakakura and Yamamoto, 1994*; *Belmonte et al., 2000*), MC calcium transients at different ages could be evoked by different neurotransmitters released during retinal waves. Thus, we next explored which transmitters modulated neuron-MC signaling at different developmental ages.

## MC calcium transients correlated with cholinergic retinal waves are mediated by muscarinic acetylcholine receptors

Our primary hypothesis is that MC calcium transients are induced by neurotransmitters released from amacrine and bipolar cells (the interneurons of the retina) during retinal waves. To assess which neurotransmitters could elicit MC calcium transients during development, we first imaged MC calcium signals in the IPL in response to periodic, focal application of agonists that could be potentially involved in the neuron-glia interaction during P7 cholinergic waves (*Figure 2A–E*). Control application of extracellular solution (artificial cerebrospinal fluid [ACSF]) did not evoke a MC response, indicating that the pressure injection itself did not evoke calcium transients through mechanical stimulation (*Figure 2C*). When adenosine tri-phosphate (ATP, 1 mM) was applied, robust calcium transients were induced that were inhibited by the P2 receptor blocker suramin (100 µM; *Figure 2D*), as seen previously in the adult retina (*Uckermann et al., 2002*; *Newman, 2004*; *Metea and Newman, 2006*). MCs also responded to acetylcholine (ACh, 1 mM; *Figure 2B,C*; *Video 2*), as described in cortical

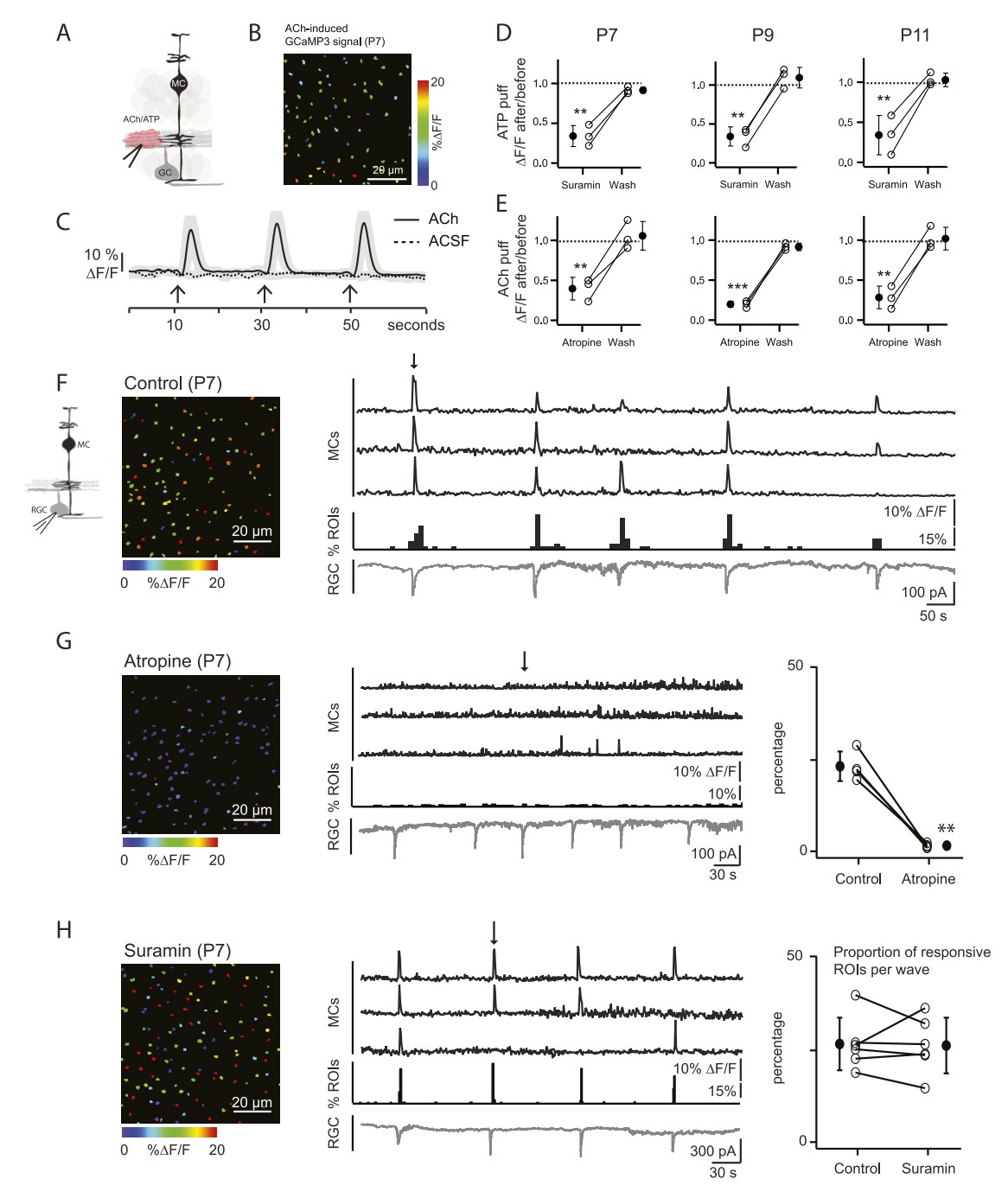

**Figure 2**. Volume release of acetylcholine (ACh) during P7 retinal waves induces calcium transients in MCs. (**A**) Diagram of retinal cross-section illustrates focal application of the agonist (pink) at the junction between the IPL and the GCL; labeling as in *Figure 1*. (**B**) XY plane of the IPL from a P7 GLASTCreER:: GCaMP3 retina showing fluorescent signals in MCs during a focal application of acetylcholine (ACh, 1 mM, 100 ms) pseudocolored as in *Figure 1C*. (**C**) Averaged MC calcium signals (ΔF/F) in a ROI evoked by a sequence of 3 focal applications of ACh (solid line; 78 ROIs from 1 retina) or artificial cerebrospinal fluid (ACSF, dashed line; 429 ROIs from 3 retinas). Black arrows indicate agonist application. Shaded areas represent standard deviation. (**D**) Averaged MC calcium signals (ΔF/F) evoked by focal application of ATP in the absence and presence of the non-specific P2 receptor blocker suramin (100 μM) at P7, P9 and P11. P7: 158 ROIs in control and 142 ROIs in suramin from 3 retinas; P9: 201 ROIs in control and 151 ROIs in suramin from 3 retinas; P11: 139 ROIs in control and 123 in suramin from 3 retinas. Black circle and error bars show mean ±SD. One-way ANOVA, Tukey *post-hoc* test ***p < 0.001; **p < 0.01. (**E**) Averaged MC calcium signals (ΔF/F) evoked by focal applications of ACh in the absence and presence of the muscarinic cholinergic receptor blocker atropine (50 μM) at P7, P9 and P11. P7: 256 ROIs in control and 396 ROIs in atropine from 3 retinas; P9: 234 ROIs in control and 264 ROIs in atropine from 3 retinas; P11: 183 ROIs in control and 121 ROIs in atropine from 3 retinas. One-way ANOVA, Tukey *post-hoc* test ***p < 0.001;

*Figure 2. continued on next page*

*Figure 2. Continued*

**p < 0.01. (**F**–**H**) Simultaneous MC calcium imaging (black traces) and RGC whole-cell voltage-clamp recording (grey trace, $V_m = -60$ mV) monitored in the same field of view from a P7 GLAST*CreER::GCaMP3* retina in control solution (**F**), in the presence of the muscarinic acetylcholine receptor blocker atropine (50 μM) (**G**), or in the presence of the non-selective P2 receptor antagonist suramin (100 μM) (**H**). Below the MC calcium transient traces are histograms showing the percentage of ROIs with responsive MCs over time. *Left*, Images show XY planes of the IPL with the MC fluorescent signals occurring at the time denoted by black arrows. *Right*, Plots summarize effects of suramin (350 ROIs in control and 390 ROIs in suramin from 6 retinas) and atropine (267 ROIs in control and 64 ROIs in atropine from 4 retinas) on the percentage of ROIs with responsive MCs per wave. Lines connect values from one experiment in control vs blocker. Black circle and error bars show mean ±SD. *t*-test **p < 0.01. See also *Video 2*.

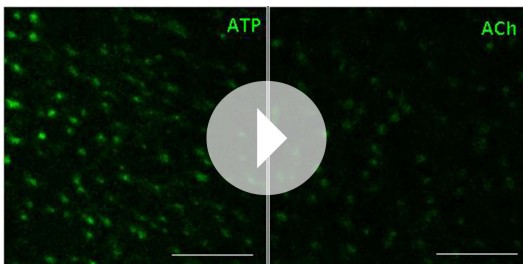

**Video 2.** Calcium transients (ΔF/F) in MCs expressing the calcium indicator GCaMP3 are shown in response to a series of focal applications of ATP or ACh (1 mM, 100 ms) at P7. Scale bars are 20 μm. White spots in video indicate when focal applications of agonist were applied. Related to *Figure 2*.

astrocytes (*Takata et al., 2011*). These ACh-evoked MC calcium transients were reduced by the muscarinic ACh receptor antagonist atropine (50 μM; *Figure 2E*). Similar ACh- and ATP-evoked MC calcium transients were also observed at P9 (during the transition from cholinergic to glutamatergic waves) and at P11 (during glutamatergic waves), indicating that MCs express multiple neurotransmitter receptors prior to eye opening (*Figure 2D,E*).

To explore the contribution of ACh and ATP to MC calcium transients that occur during P7 cholinergic waves, simultaneous two-photon laser scanning microscopy and whole-cell RGC recordings were performed. MC calcium transients (*Figure 2F*, top black traces) were correlated with RGC compound excitatory postsynaptic currents (EPSCs) that are known to be mediated by the volume release of ACh and the activation of neuronal nicotinic ACh receptors (AChRs) on RGCs (*Figure 2F*, bottom grey traces) (*Ford et al., 2012*). Interestingly, this ACh is released by cholinergic amacrine cells that also release ATP (*Duarte et al., 1999*). Thus, we tested the effect of AChR and ATP receptor blockers on wave-induced MC calcium transients. Although the wave-induced transients were blocked by the muscarinic ACh receptor (mAChR) antagonist atropine (50 μM; *Figure 2G*), they were not blocked by the ATP receptor antagonist suramin (100 μM; *Figure 2H*). Therefore, we conclude that at P7, MC calcium transients are evoked by volume release of ACh, but not by ATP, and these transients are mediated by mAChRs on MCs.

## MC calcium transients correlated with glutamatergic retinal waves are limited by the extent of glutamate spillover and are mediated by AMPA receptors

At P9, circuits mediating retinal waves change from cholinergic to glutamatergic (*Blankenship and Feller, 2010*), and glutamatergic synapses between bipolar cells and their postsynaptic targets first appear (*Hoon et al., 2014*). Thus, at P9 cholinergic and/or glutamatergic transmission could activate MC calcium transients. To determine the involvement of cholinergic transmission in MC signaling at this stage of development, we examined their response to focal application of ACh. Short applications of ACh induced calcium transients that were blocked by the mAChR blocker atropine at P9 (*Figure 2E*), but bath application of atropine did not change the percentage of wave-induced MC calcium transients (57 ± 12.6% in control and 60.2 ± 7.8% in atropine; p = 0.54, n = 4). These results indicate that cholinergic signaling does not mediate neuron-MC signaling at this age. We then determined the role of glutamate in triggering these responses. Application of exogenous L-glutamate reliably evoked MC calcium transients (*Figure 3A–C*; *Video 3*). These responses were not blocked by atropine, ruling out the possibility that this agonist acted indirectly by stimulating release of ACh from amacrine cells. L-glutamate-evoked MC calcium transients were significantly reduced by a cocktail of ionotropic glutamate receptor antagonists (iGluR: 20 μM AMPA/kainate

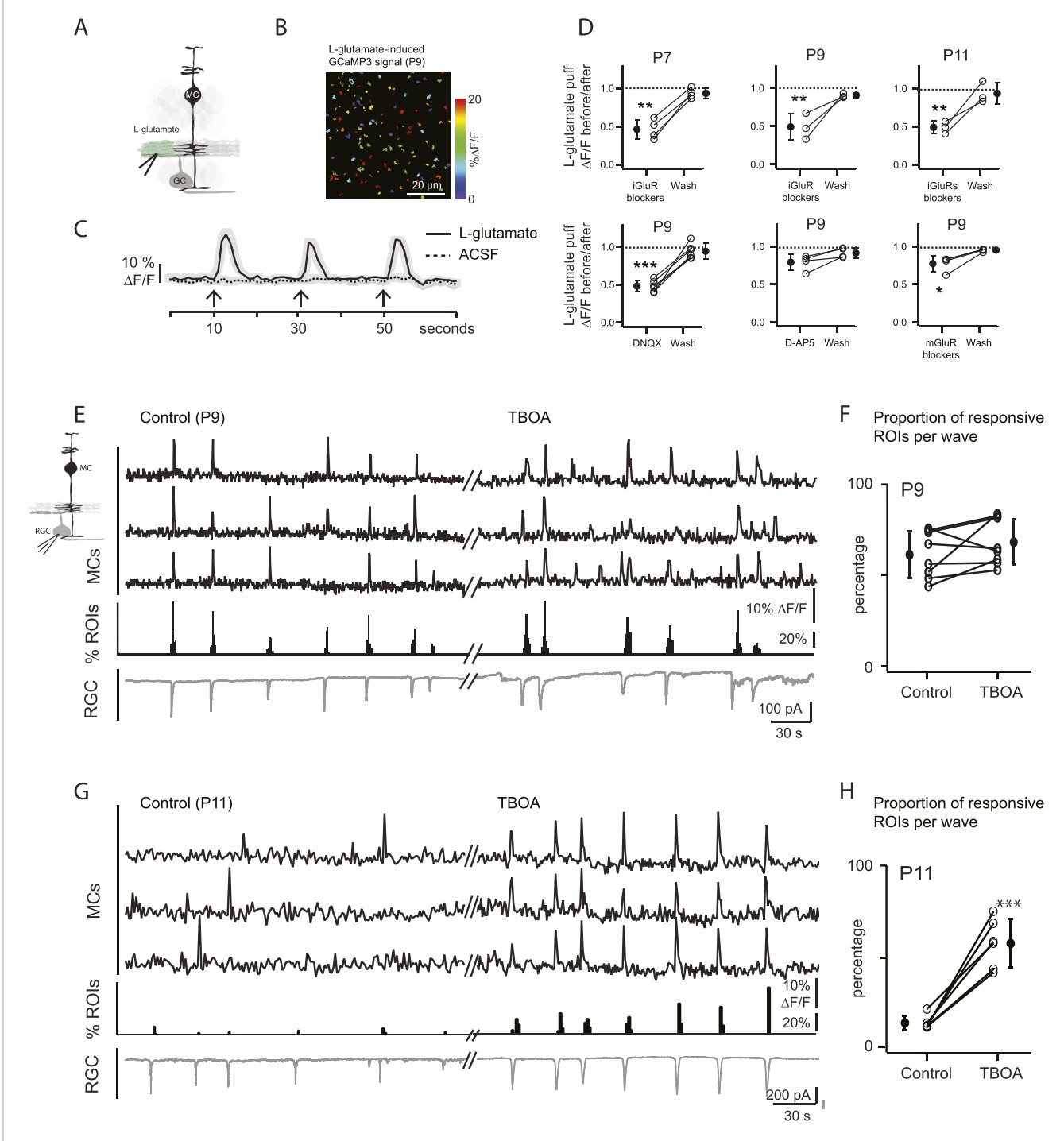

**Figure 3**. Wave-induced glial signaling is limited by extent of glutamate spillover. (**A**) Diagram of retinal cross-section illustrates focal application of the agonist (green) at the junction between the IPL and the GCL; labeling as in *Figure 1*. (**B**) XY plane of the IPL from a P9 GLAST*CreER::GCaMP3* retina shows fluorescent signals in MCs during short-application of L-glutamate (1 mM, 100 ms) pseudocolored as in *Figure 1C*. (**C**) Averaged MC calcium signals (ΔF/F) in a ROI evoked by a sequence of focal applications of L-glutamate (solid line; 145 ROIs from 3 retinas) or ACSF (dashed line; 78 ROIs from 1 retina). Black arrows represent agonist application. Shaded areas represent standard deviation. (**D**) *Top*, Averaged MC calcium signals (ΔF/F) evoked by focal applications of L-glutamate in the absence and presence of a cocktail of ionotropic glutamate receptor blockers (iGluR blockers; 20 μM DNQX and 50 μM D-AP5) at P7 (462 ROIs from 3 retinas), P9 (708 ROIs from 4 retinas) and P11 (1342 ROIs from 5 retinas). *Bottom*, Averaged MC calcium signals (ΔF/F) evoked by focal application of L-glutamate in the absence and presence of the ionotropic AMPA glutamate receptor blocker DNQX (20 μM; 1311 ROIs from 7 retinas), the ionotropic NMDA glutamate receptor blocker D-AP5 (50 μM; 907 ROIs from 4 retinas) and a cocktail of metabotropic glutamate receptor blockers (mGluR blockers; 8 μM

*Figure 3. continued on next page*

*Figure 3. Continued*

LY341495, 100 µM MCPG and 100 µM MTEP; 937 ROIs from 4 retinas) at P9. One-way ANOVA, Tukey *post-hoc* test ***p < 0.001; **p < 0.01 and *p < 0.05. (**E–H**) Simultaneous MC calcium imaging (black traces) and RGC whole-cell voltage-clamp recording (grey trace, $V_m = -60$ mV) monitored in the same field of view at P9 (**E**) and P11 (**G**) in control solution and in the presence of 25 µM DL-TBOA. Below calcium traces are histograms showing the percentage of ROIs with responsive MCs over time. Plots summarize DL-TBOA effects on the percentage of ROIs with responsive MCs per retinal wave at P9 (**F**; 3027 ROIs in control and 3549 ROIs in DL-TBOA from 8 retinas) and P11 (**H**; 872 in control and 1342 in DL-TBOA from 6 retinas). Lines connect values from one experiment in control vs DL-TBOA. Black circle and error bars show mean ±SD. *t*-test, ***p < 0.001. See also *Video 3* and *Video 1*.

receptor antagonist DNQX, 50 µM NMDA receptor antagonist D-AP5; *Figure 3D*), but were only slightly reduced by a mixture of metabotropic glutamate receptor antagonists (mGluR: 8 µM LY341495, 100 µM MCPG, 100 µM MTEP; *Figure 3D*). Blocking AMPA receptors produced substantial but incomplete inhibition of the L-glutamate-evoked response, whereas blocking NMDA receptors had no significant effect. These results suggest that glutamate elicits MC calcium transients primarily through activation of AMPA receptors, although a modest contribution of mGluRs is also observed. Notably, the reduction of L-glutamate-evoked calcium transients by glutamate receptor blockers occurred at all ages, not just during the period when glutamatergic waves predominate, indicating that MCs express glutamate receptors throughout this period of development.

To study MC calcium transients evoked by glutamatergic waves, we combined two-photon laser scanning microscopy with whole-cell recordings from RGCs at P9 and P11. At P9, wave-induced MC calcium transients were correlated with compound EPSCs in RGCs (*Figure 3E,F*) that are mediated by release of glutamate from bipolar cells and by activation of ionotropic glutamate receptors on RGCs (*Bansal et al., 2000*; *Wong et al., 2000*; *Zhou and Zhao, 2000*). However, at P11, wave-induced MC calcium transients were dramatically reduced (*Figures 1E, 3G,H*) although robust glutamate spillover occurs at this age (*Blankenship et al., 2009*; *Firl et al., 2013*) and exogenous glutamate induced MC calcium transients (*Figure 3B–D*).

We hypothesize that this reduction of MC activity is a consequence of the upregulation of glutamate transporters at P11 (*Pow and Barnett, 2000*; *Pannicke et al., 2002*), which may limit the amount of glutamate that reaches MC membranes to induce calcium transients. To test this hypothesis, we blocked glutamate transporters by bath applying DL-TBOA (25 µM), an agent previously shown to increase extracellular glutamate levels (*Blankenship et al., 2009*; *Firl et al., 2013*). At P11, DL-TBOA drastically increased the percentage of responsive MCs from $13 \pm 2.2\%$ to $61 \pm 4.3\%$ (*Figure 3G,H*; 872 ROIs in control and 1342 ROIs in DL-TBOA from 6 retinas). Interestingly, at P9, DL-TBOA did not significantly change the percentage of responsive MCs, increasing only slightly from $48 \pm 4.5\%$ to $58 \pm 10.0\%$ (*Figure 3E,F*; 3027 ROIs in control and 3549 ROIs in DL-TBOA from 8 retinas). We conclude that at ages earlier than P11, transporters do not strongly regulate glutamate spillover, but by P11, they mediate significant glutamate uptake, decreasing the amount of glutamate that reaches MC receptors.

We expressed the membrane-bound intensity-based glutamate sensing fluorescent reporter iGluSnFR (*Marvin et al., 2013*) either in MCs or neurons (*Borghuis et al., 2013*) to test directly if extracellular glutamate was reaching the membrane of MCs during retinal waves. The iGluSnFR signal is proportional to the amount of glutamate reaching the membrane (*Marvin et al., 2013*); thus, the lack of fluorescence in the INL where MC cell bodies are located indicates that during early development there is little glutamate release outside of the IPL.

To detect glutamate signals during retinal waves, we simultaneously performed two-photon imaging of iGluSnFR signals in the IPL and whole-cell voltage clamp recordings from RGCs (*Figure 4*; *Figure 4—figure supplement 1*).

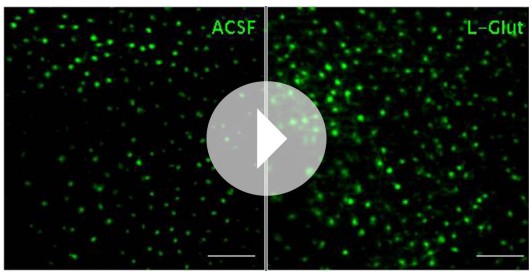

**Video 3.** Calcium transients (ΔF/F) in MCs expressing the calcium indicator GCaMP3 are shown in response to a series of focal applications (100 ms) of ACSF or L-glutamate (1 mM) at P9. Scale bars are 20 µm. White spots in video indicate when focal applications of agonist were applied. Related to *Figure 3*.

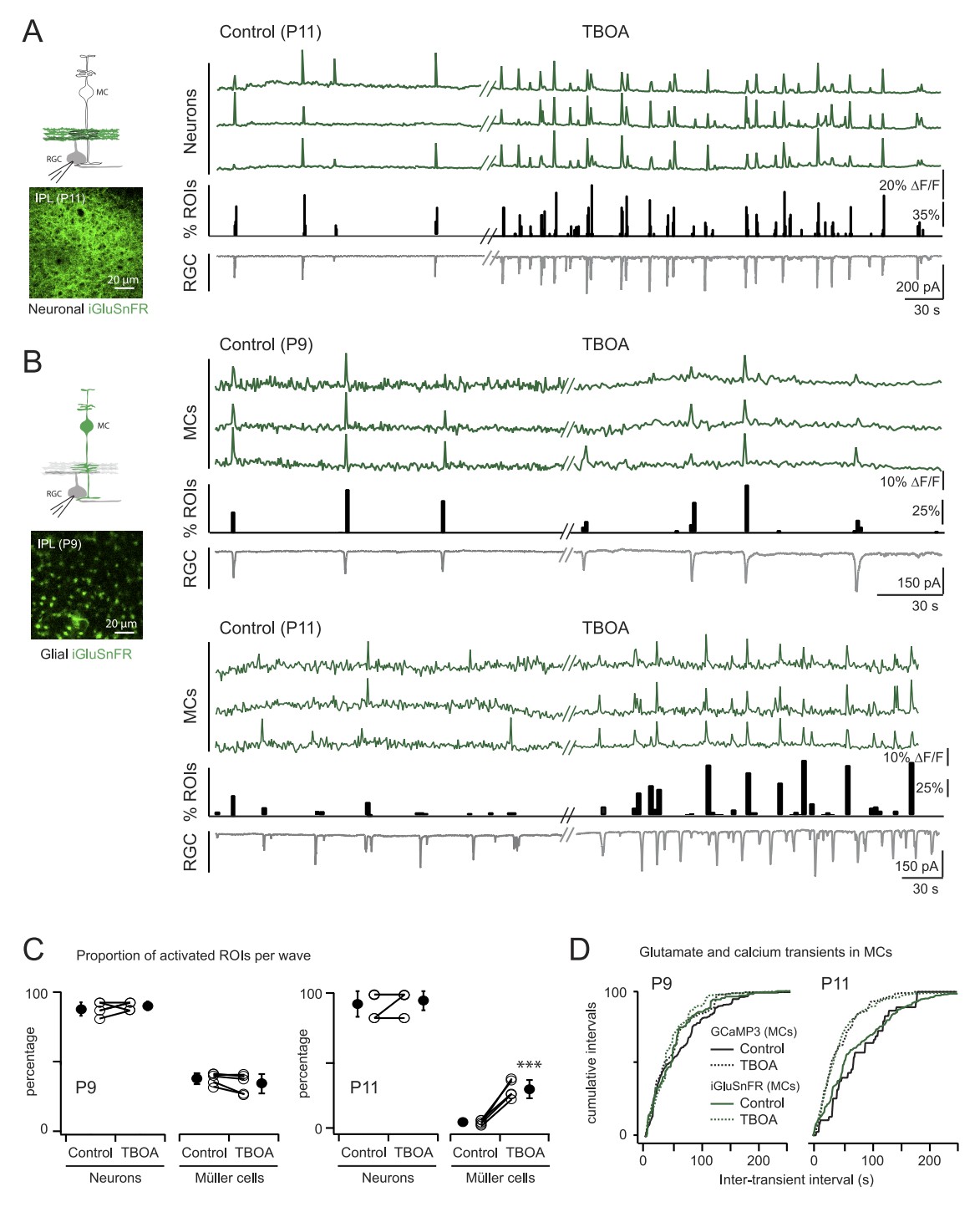

**Figure 4**. Glutamate released during neuronal waves reaches MC membrane at P9 but not at P11. (**A**) *Left,* Diagram of a retinal cross-section illustrates neuronal expression of iGluSnFR (green) in the IPL and simultaneous voltage-clamp recording of a RGC (grey). XY plane of the IPL shows iGluSnFR expression in neuronal membranes at P11. *Right,* Simultaneous imaging of AAV9-2YF-hSynapsin-iGluSnFR signals in neuronal membranes (green traces) and whole-cell voltage-clamp recording of a RGC (grey trace, $V_m = -60$ mV) monitored in the same field of view at P11 in control and in the presence of 25 μM DL-TBOA. Above the whole-cell voltage-clamp trace are histograms showing the percentage of ROIs within a neuronal iGluSnFR signal. (**B**) *Left,* Diagram of a retinal cross-section illustrates glial expression of iGluSnFR (green) and simultaneous voltage-clamp recording of a RGC (grey). XY plane of the IPL shows iGluSnFR expression in MCs. *Right,* Simultaneous imaging of ShH10-CMV-iGluSnFR signals in MCs (green traces) and whole-cell voltage-clamp recording of a RGC (grey trace, $V_m = -60$ mV) monitored in the same field of view at P9 and P11 in control and in the presence of 25 μM DL-TBOA. Above each whole-cell voltage-clamp trace are histograms showing the percentage of ROIs with glial iGluSnFR signals in response to retinal waves.
*Figure 4. continued on next page*

*Figure 4. Continued*

(**C**) Plot summarizes DL-TBOA effects on the participation of neuronal (160 ROIs from 5 retinas at P9 and 160 ROIs from 5 retinas at P11) and MC (1023 ROIs from 5 retinas at P9 and 1201 ROIs from 5 retinas at P11) ROIs per retinal wave. Lines connect values from one experiment in control vs DL-TBOA. Black circle and error bars are mean ±SD. *t*-test ***p < 0.001. (**D**) Cumulative probability distribution of inter-transient intervals of iGluSnFR (green traces) and GCaMP3 (black traces) signals in MC ROIs at P9 and P11. Control in solid lines and DL-TBOA in dashed lines. See also *Figure 4—figure supplement 1* and *Videos 4, 5*.

The following source data and figure supplement are available for figure 4:

**Source data 1**. Cumulative probability distributions of inter-transient intervals of iGluSnFR and GCaMP3 signals in MC ROIs for each experiment at P9 and P11, and in absence or in presence of DL-TBOA.

**Figure supplement 1**. Methodological tools to study neuron-glia interaction mediated by glutamate spillover in whole mount retinas.

At both P9 and P11, the processes of neurons that expressed iGluSnFR exhibited large glutamate transients that propagated throughout the field of view with each retinal wave (*Figure 4A,C*; 160 ROIs from 5 retinas at P9 and 160 ROIs from 5 retinas at P11; *Video 4*), consistent with the key role played by glutamate in inducing retinal waves at these ages. In contrast, although many MCs expressing iGluSnFR displayed glutamate transients during P9 waves (38 ± 4%, 1023 ROIs from 5 retinas, *Figure 4B,C*), the percentage of responsive MCs decreased dramatically by P11 (6 ± 2%, 269 ROIs from 5 retinas at P11; *Figure 4B,C*). Inhibiting glutamate transporters with DL-TBOA (25 μM) increased the percentage of responsive MCs at P11 (31 ± 7%, 822 ROIs from 5 retinas; *Video 5*) and reduced the inter-event intervals for MC calcium and glutamate transients (*Figure 4D*; *Figure 4—source data 1*), indicating that glutamate uptake limits spillover-induced activation of MCs at this age. DL-TBOA exerted a minimal effect on MC calcium and glutamate transients at P9 (*Figure 4B–D*). These findings corroborate the hypothesis that the lower expression of glutamate transporters at P9 enables robust entrainment of MCs during retinal waves.

Several lines of evidence support the hypothesis that MC calcium transients depend upon the direct activation of MC glutamate receptors during the period of glutamatergic retinal waves. First, MCs are known to express AMPARs (*Wakakura and Yamamoto, 1994*; *Sullivan and Miller, 2010*). Second, glutamate-induced MC calcium transients depend upon the activation of ionotropic AMPARs (*Figure 3D*). Third, the frequencies of glutamate and calcium transients in MCs are indistinguishable, indicating that MC calcium transients are driven by glutamate reaching the MC membranes (*Figure 4D; Figure 4—source data 1*). Establishing that wave-induced MC calcium transients are mediated by ionotropic glutamate receptors on MCs requires selective inhibition of these receptors; however, this approach is complicated by the fact that depolarization of RGCs during retinal waves also depends on AMPAR activation. Since the AMPAR antagonist DNQX may have an effect on glutamate release from bipolar cells, we tried to elucidate its effects in this circuit. In the presence of DNQX, we simultaneously monitored EPSCs from RGCs with whole-cell recordings and glutamate release from bipolar cells with neuronal iGluSnFR imaging (*Figure 5A,B*). As described previously (*Blankenship et al., 2009*), DNQX (20 μM) significantly decreased the amplitude of the postsynaptic current in RGCs. However, it did not change the amount of wave-induced glutamate release from bipolar cells monitored using iGluSnFR (*Figure 5A*). With this knowledge, we used DNQX to determine the role of AMPARs in wave-induced MC calcium transients. These experiments were conducted in the presence of DL-TBOA to maximize the

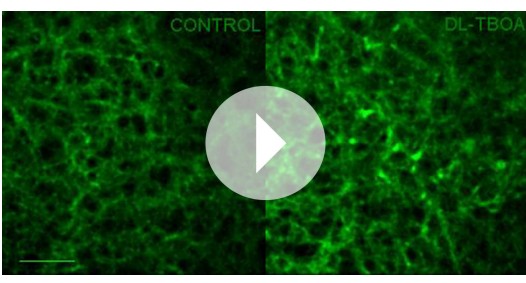

**Video 4.** Wave-induced fluorescence changes of the glutamate sensor iGluSnFR expressed in RGCs and amacrine cells in a P11 mouse retina are shown in the presence of the glutamate uptake blocker DL-TBOA (25 μM). Electrophysiological recordings confirmed that glutamate signals were correlated with RGC activity during retinal waves. Scale bars are 10 μm. Related to *Figure 4*.

**Video 5.** Wave-induced fluorescence changes of the intensity-based glutamate sensing fluorescent reporter iGluSnFR expressed specifically in MCs in a P11 mouse retina are shown in the presence of the glutamate uptake blocker DL-TBOA (25 μM). Electrophysiological recordings confirmed that glutamate signals were correlated with RGC activity during retinal waves. Scale bars are 20 μm. Related to *Figure 4*.

amount of glutamate spillover. We found that DNQX significantly decreased the percentage of responsive MCs during waves (*Figure 5C*, *Figure 5—figure supplement 1A*), whereas the NMDAR antagonist D-AP5 (50 μM) had no significant effect on MC activity (*Figure 5—figure supplement 1B,C*; 703 ROIs in control and 656 ROIs in D-AP5 from 2 retinas). Our results strongly suggest that glutamate acts directly on MC AMPARs to increase intracellular calcium. However, it will be necessary to selectively delete AMPARs from MCs to establish their contribution to MC calcium signaling during this period of development.

Previous studies have shown that mGluRs contribute to neuron-astrocyte signaling in the brain, particularly during early development (*Grosche et al., 1999*; *Kirischuk et al., 1999*).

To determine the extent that mGluRs are involved in wave-induced MC calcium transients in the retina we performed two experiments. First, we conducted simultaneous whole-cell recordings from RGCs and two-photon calcium imaging of MC activity in the presence of mGluR antagonists (8 μM LY341495, 100 μM MCPG, 100 μM MTEP). In these conditions, there was a small, but not significant, decrease in the percentage of responsive MCs at P9 (*Figure 6D*), consistent with the contribution of mGluRs to MC calcium transients evoked by exogenous L-glutamate (*Figure 3B–D*). Second, we imaged the spontaneous activity of MCs in retinas isolated from mice that lack $IP_3R2$ ($IP_3R2$-KO mice), which has been shown to be required for metabotropic receptor-induced calcium signaling in adult MCs (*Lipp et al., 2009*), using a fluorescent calcium dye (*Figure 6A,B*). The density of MC labeling was not dramatically different between GCaMP3 expressing mice vs MCs loaded with organic dyes, although this was not compared quantitatively. Wave-like calcium transients persisted in $IP_3R2$-KO MCs and a similar percentage of MCs participated in these events (*Figure 6C,D*) indicating that $IP_3$–dependent calcium signaling contributes minimally to MC calcium transients during retinal waves. Together, these findings suggest that volume release of glutamate during retinal waves evokes MC calcium transients primarily by activating AMPA receptors on MCs.

## Discussion

In this study we elucidated the spatio-temporal characteristics and molecular pathways by which neurons and MCs communicate in the developing retina. We found that retinal waves were correlated with calcium transients in a large proportion of MCs, demonstrating that spontaneous activity encompasses both neuronal and glial networks at this age. Our physiological results indicate that MCs express multiple neurotransmitter receptor types throughout this period, allowing neuronal control of MC activity to be sustained as neuronal synchronization shifts from being dependent on cholinergic to glutamatergic signaling. The proportion of MCs that responded to waves with calcium transients decreased significantly from P9 to P11, despite the presence of retinal waves. This age-dependent decline in MC activation during retinal waves was caused by an increase in glutamate transporter expression, which restricted glutamate spillover from nearby synapses limiting activation of MC AMPARs. These data indicate that MCs in the developing retina use multiple signaling pathways to participate in spontaneous, correlated activity with neurons during a crucial period of retinal development.

Many different signaling mechanisms have been implicated in inducing calcium transients in astroglial cells throughout the peripheral and central nervous system (*Volterra et al., 2014*; *Haydon and Nedergaard, 2015*; *Nimmerjahn and Bergles, 2015*). The robustness and mechanisms of signaling vary tremendously, even at the same synapses, with age, preparation, and state of arousal. Our data indicate that MCs in the retina prior to eye opening are capable of responding to different neurotransmitters with increases in intracellular calcium (*Figures 2A, 3A*). Neurotransmitter-evoked calcium transients were observed in the stalks and lateral processes of MCs that transverse the IPL during a time when there is robust synaptogenesis. These multiple receptors allow MCs to detect

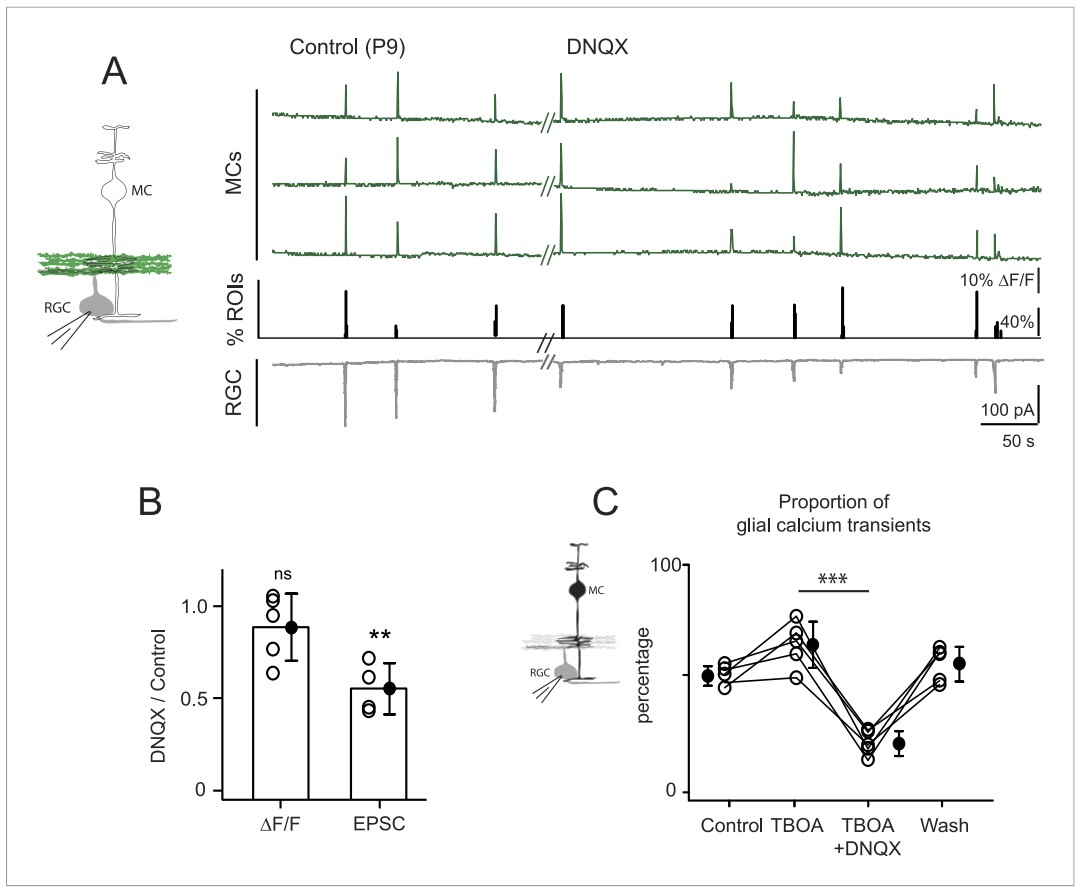

**Figure 5**. Neuron-glia interaction is mediated by ionotropic AMPA glutamate receptors. (**A**) *Left*, Diagram of a retinal cross-section illustrates neuronal expression of iGluSnFR (green) in the IPL and simultaneous voltage-clamp recording of a RGC (grey). *Right*, Simultaneous imaging of AAV9-2YF-hSynapsin1-iGluSnFR signals in neuronal membranes (green traces) and whole-cell voltage-clamp recording of a RGC (grey trace, $V_m = -60$ mV) monitored in the same field of view at P9 in control and in the presence of 20 µM DNQX. Above each whole-cell voltage-clamp trace are histograms showing the percentage of ROIs with responsive neuronal iGluSnFR signals. (**B**) Graph summarizes the effect of DNQX on the change in volume release of glutamate (ΔF/F) and on the change in amplitude of the RGC excitatory postsynaptic currents. Each open circle plots the value from one experiment. Note, DNQX does not modify the amount of glutamate released from bipolar cells. Black circle and error bars are mean ± SD. *t*-test, **p < 0.01. (**C**) *Left,* Diagram of a retinal cross-section illustrates glial expression of GCaMP3 (black) and simultaneous voltage-clamp recording of a RGC (grey). *Right*, Graph summarizes the effect of DNQX on the percentage of ROIs with responsive glia at P9 in the presence of DL-TBOA (25 µM). Lines connect values from one experiment in control (974 ROIs), DL-TBOA (1154 ROIs), DL-TBOA+DNQX (397 ROIs) and wash (987 ROIs). Data collected from 5 retinas. Black circle and error bars are mean ±SD. One-way ANOVA, Tukey *post-hoc* test ***p < 0.001. See also *Figure 5—figure supplement 1*.

The following figure supplement is available for figure 5:

**Figure supplement 1**. The AMPA receptor antagonist DNQX, but not the NMDA receptor antagonist D-AP5, decreases neuron-glia interaction at P9.

neurotransmitter released during retinal waves whether it is ACh or glutamate, thereby extending the developmental period over which neuron-glia signaling occurs.

The neuron-glia signaling we observed between neurons and MCs in the retina during glutamatergic waves is similar to signaling that occurs in Bergmann glial cells (BGs), a type of radial astroglial cell found in the cerebellum. BGs express calcium permeable AMPARs that are activated by glutamate released at parallel and climbing fiber synapses (*Bergles et al., 1997*; *Clark and Barbour, 1997*; *Piet and Jahr, 2007*), raising the possibility that AMPAR expression may be a conserved

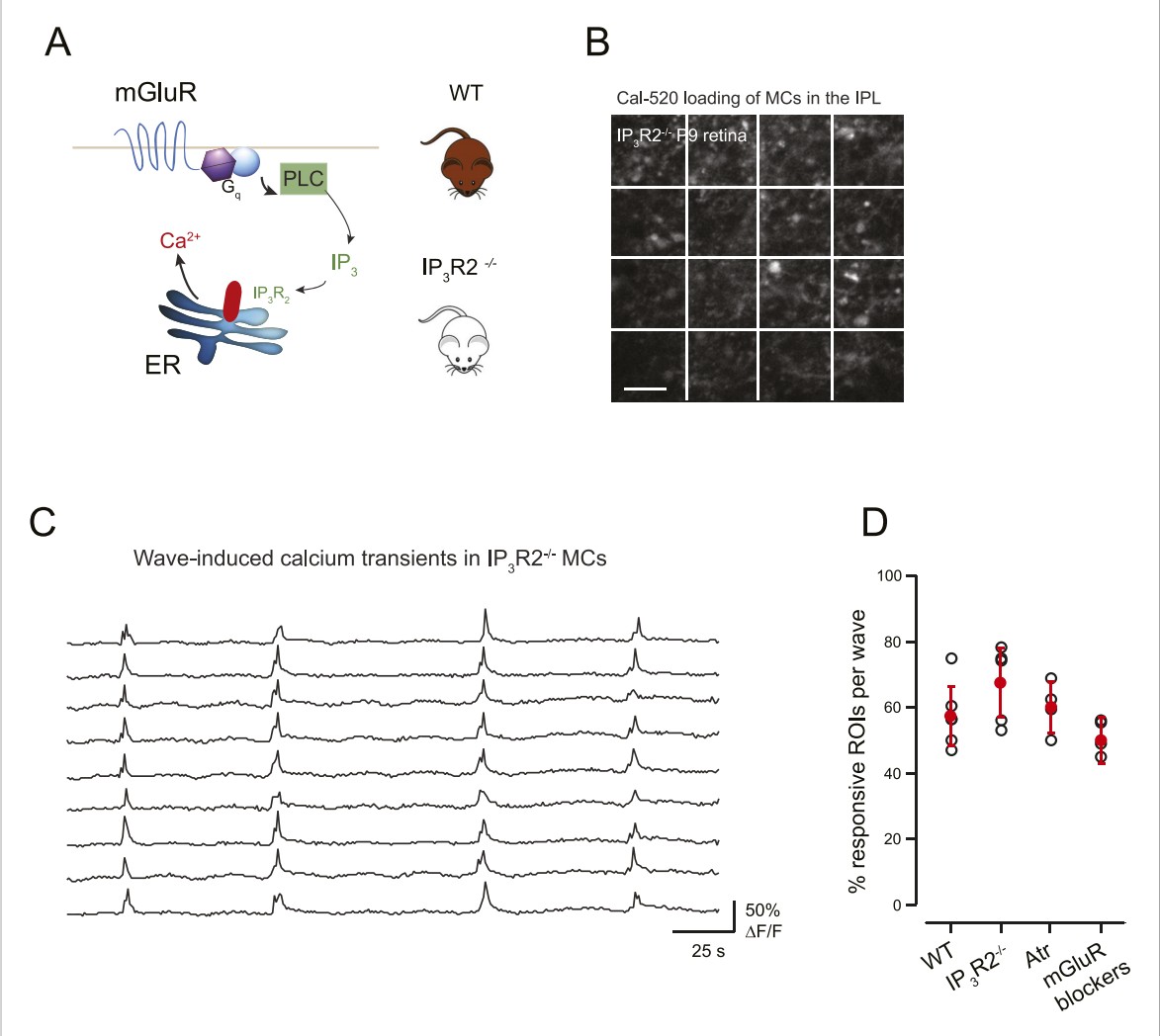

**Figure 6**. mGluR and mAChRs contribute minimally to wave-induced MC calcium transients at P9. (**A**) Schematic of mGluR-mediated pathway for increases in intracellular calcium in Müller glial cell. (**B**) XY plane of the IPL shows Cal-520 loading from an IP3R2-KO P9 retina. The whole field of view was divided in 16 ROI represented by the white rectangles. (**C**) Example traces of spontaneous MC calcium signals (ΔF/F) in IP3R2-KO retinas. (**D**) Summary of the effects of genetic and pharmacological manipulations of GPCR signaling in MCs using IP3R2-KO mice, mAChR antagonist (50 μM atropine), and mGluR antagonists (8 μM LY341495, 100 μM MCPG, 100 μM MTEP).

properties of radial-type astroglial cells. Although the role of AMPARs in BG activity is still controversial (*Müller et al., 1996*; *Iino et al., 2001*; *Nimmerjahn et al., 2009*), in vivo studies indicate that calcium signaling in BGs during locomotion depends on the activation of AMPARs (*Nimmerjahn et al., 2009*) and disruption of this signaling (*Iino et al., 2001*; *Saab et al., 2012*), results in prolonged synaptic events that are associated with altered motor behavior and motor learning. Moreover, the mechanism of interaction between neurons and MCs we observed during cholinergic retinal waves is similar to previous findings in visual cortex and hippocampus, where cholinergic neuron stimulation leads to astrocyte calcium transients via activation of muscarinic acetylcholine receptors and require IP3R2 (*Chen et al., 2012*; *Navarrete et al., 2012*).

Whether neuronal activity during retinal waves guides the development of MCs is still unclear. MCs exhibit dramatic morphological changes during the period from P7–P11 as their fine processes grow laterally into the IPL and ensheath synapses. Ensheathment of synapses by astrocytes has been shown to promote functional isolation of synaptic signaling by creating a barrier to diffusion of neurotransmitters and by placing neurotransmitter transporters near sites of release. It is possible

that ablation of AMPARs in MCs would result in retraction of the lateral processes away from synapses, as observed in BGs after inhibition of calcium permeable AMPARs (*Iino et al., 2001*) or genetic ablation of GluA1 and GluA4 (*Saab et al., 2012*). Our findings in the developing retina differ from studies performed in the adult retina, in which glutamate does not evoke calcium transients in MCs (*Newman and Zahs, 1997*; *Newman, 2005*), suggesting that glutamaterigic signaling through AMPARs in these cells may be limited to the early stages of synapse formation and refinement.

Astrocytes have also been shown to play critical roles in promoting the refinement and maturation of neural circuits (*Stevens, 2008*; *Clarke and Barres, 2013*; *Chung et al., 2015*) by secreting synaptogenic factors, such as thrombospondin, hevin, protocadherins, BDNF, and TGF-beta, which are critical for synapse formation between RGCs and their central targets in the brain (reviewed in *Chung and Barres, 2012*; *Clarke and Barres, 2013*). In addition, astrocytes play an important role in refining nascent circuits in the visual system by engulfing unused or inappropriate synapses (*Chung et al., 2015*). Our findings indicate that robust signaling occurs between retinal neurons and MCs between the ages P7 and P11, a time period when there is robust synaptogenesis in the retina (*Hoon et al., 2014*). The close lineage relationship and functional similarities between astrocytes and MCs increase the possibility that MCs perform similar roles in the retina, particularly in regions such as the IPL where they are the sole astroglial cell.

Previous studies have shown that changes in intracellular calcium can trigger the release of so-called gliotransmitters such as ATP, glutamate and D-serine from astrocytes that can influence the activity of surrounding neurons (*Araque et al., 2014*; *Newman, 2015*). In the retina, activation of AMPARs promotes the release of D-serine, a co-agonist of NMDA receptors in ganglion cells, that has been shown to play an important role in synaptic plasticity (*Stevens, 2008*; *Sullivan and Miller, 2010*), raising the possibility that MC activity induced during retinal waves participates in synaptic refinement through release of gliotransmitters and/or synaptogenic factors. Selective manipulation of MC calcium signaling during this period of remarkable refinement will help to further define their roles in the development of distinct retinal circuits.

## Materials and methods

### Animals
*GLASTCreER::GCAMP3* mice were generated by cross breeding GLAST-*CreER* BAC transgenic mice [Tg(Slc1a3-cre/ERT)1Nat/J] (*de Melo et al., 2012*) to *ROSA26-lsl-GCaMP3* reporter mice (*Paukert et al., 2014*). GLAST*CreER::*tdTomato mice were generated by crossing GLAST*CreER*BAC mice to mice that ubiquitously express tdTomato preceded by a loxP-flanked stop cassette (B6;129S6-*Gt(ROSA)26Sor*$^{tm9(CAG-tdTomato)Hze}$/J, from The Jackson Laboratory, Bar Harbor, ME). The gene targeting and generation of IP$_3$R2 knock-out mice have been previously described (*Li et al., 2005*). All animal procedures were approved by the University of California (UC) Berkeley and Johns Hopkins University's Animal Care and use Committees and conformed to the NIH Guide for the Care and Use of Laboratory Animals, the Public Health Service Policy, and the SFN Policy on the Use of Animals in Neuroscience Research.

### Viral expression strategies and constructs
Intraperitoneal injection of 0.5 mg of 4-hydroxytamoxifen (50:50 E and Z isomers, Sigma-Aldrich, St Louis, MO) at 4 and 2 days before each experiment reliably induced the expression of GCaMP3 in MCs by P5, P7 or P11. For the glutamate experiments, the intensity-based glutamate sensing fluorescent reporter iGluSnFR (*Borghuis et al., 2013*; *Marvin et al., 2013*) was specifically expressed either in MCs or in retinal neurons (ganglion and amacrine cells). Expression of iGluSnFR in MCs was accomplished by intravitreally injecting C57BL/6J mice (The Jackson Laboratory, Bar Harbor, ME) of either sex at P4 with a capsid-modified adeno-associated virus incorporating a CMV promoter driving expression of the iGluSnFR cDNA (ShH10-CMV-iGluSnFR; 1 µl) (*Klimczak et al., 2009*; *Dalkara et al., 2011*). Expression of iGluSnFR in neurons was performed via tail vein injection at P2 of adeno-associated virus serotype 9 carrying 2 tyrosine mutations (*Dalkara et al., 2011*) under control of the human synapsin-1 promoter (AAV9-2YF-hSynapsin1-iGluSnFR, 10 µl) from P2 C57BL/6J mice. All AAV vectors were produced according to the methods described in *Flannery and Visel (2013)*.

## Calcium-dye loading of MCs

Calcium indicator Cal-520-AM (AAT Bioquest, Sunnyvale, CA; *Figure 6*) was bath loaded into MCs using methods analogous to other organic dyes (*Newman and Zahs, 1997*; *Bansal et al., 2000*). Isolated retinas were incubated with Cal-520 (10 µM) prepared in ACSF containing 1% DMSO and 0.02% pluronic acid for 30 min in an oxygenated chamber at 32°C. The specificity of calcium signals in MCs was achieved by using the AM version of the organic dye that restricts loading to glial cells in developing retina and by two-photon imaging in the IPL. Specificity of loading into MCs was confirmed by comparison with images obtained from GLAST*CreER::R26-lsl-tdTomato* mice (GLAST*CreER::tdTomato* mice).

The evidence that expression was restricted to MCs was based on morphology. MCs have a distinct morphology, in which they traverse the entire retina with lateral processes expanding from their stalks at the level of IPL and OPL. Our two-photon and confocal images of tdTomato immunofluorescence from GLAST*CreER::tdTomato* mice confirmed these morphological features (*Figure 1B*).

## Retinal preparation

The animals were anesthetized with isoflurane inhalation and decapitated. After enucleation of the eyes, the retinas were dissected in oxygenated (95% $O_2$–5% $CO_2$) ACSF (containing [in mM] 119.0 NaCl, 26.2 $NaHCO_3$, 11 glucose, 2.5 KCl, 1.0 $K_2HPO_4$, 2.5 $CaCl_2$, and 1.3 $MgCl_2$). The isolated retinas were then mounted ganglion cell side up on filter paper (Millipore, Billerica, MA) and transferred into the recording chamber of an upright microscope for simultaneous imaging and electrophysiological recording.

## Immunohistochemistry

Retinas were fixed in 4% paraformaldehyde and cut into 120 µm sections with a vibratome. Sections were washed in PBS (10 min) and then a blocking solution (2% donkey serum, 1% bovine serum albumin, 0.2% Triton X-100; 45 min at room temperature). Sections were then incubated in primary antibodies at 4°C overnight (1:200 guinea-pig anti-VGLUT1, Chemical International, Temecula, CA). Retinas were washed in blocking solution (3 times, 10 min). The sections were subsequently incubated in a secondary antibody (1:200 Alexa fluor 647 anti-guinea-pig, Invitrogen, Grand Island, NY; 45 min) and then washed in blocking solution (3 times, 10 min). Sections were mounted on slides in anti-fade medium containing 4′,6-diamidino-2-phenylindole and imaged with a Zeiss LSM 780 confocal microscope. Note, the labeling of MC somas from the GLAST*CreER::tdTomato* mice is more apparent than in the GLAST*CreER::GCaMP3* mice because it was visualized via immunofluorescence, which significantly improved signal-to-noise. In addition, tdTomato confocal images were acquired from retinal slices, while GCaMP3 two-photon images were acquired from whole mount flat retinas, where signal strength was reduced as a function of depth.

## Two-photon imaging

Two-photon imaging of neurons and MCs in the IPL was performed using either a custom-modified two-photon microscope (Fluoview 300, Olympus America, Melville, NY) or a custom-built two-photon microscope. Time series images were acquired using Olympus 60×, 1 NA, LUMPlanFLN objectives, and two-photon excitation of GCAMP3 was evoked with an ultrafast pulsed laser (Chameleon Ultra, Coherent, Santa Clara, CA) tuned to 920 nm on both microscopes. For imaging Cal520 (*Figure 6*) the laser was tuned to 820 nm. The microscope was controlled by Fluoview Viewer software or ScanImage software (version 3.8, www.scanimage.org). Images (256 × 256 pixels) were acquired at 0.74 or 1.7 Hz at 2 or 4 ms/line. Scan parameters were [pixels/line × lines/frame (frame rate in Hz)]: [256 × 256 (0.74 − 1.7)], at 2–4 ms/line. Line scans were obtained at 300 Hz and down-sampled to 30 Hz for presentation in *Figure 1—figure supplement 1*.

## Electrophysiological recordings

Whole-cell voltage clamp recordings were made from whole-mount retinas continuously superfused in oxygenated ACSF (32–34°C) at a rate of 2–4 ml/min. Retinas were visualized under infrared illumination (870 nm). Voltage-clamp recordings from somas of ganglion cells (holding potential of −60/−65 mV) were obtained using glass microelectrodes of 4–5 MΩ (PC-10 pipette puller; Narishige, East Meadow,

NY) filled with an internal solution containing (in mM): 110 CsMeSO$_4$, 2.8 NaCl, 4 EGTA, 5 TEA-Cl, 4 adenosine 5′-triphosphate (magnesium salt), 0.3 guanosine 5′-triphosphate (trisodium salt), 20 HEPES and 10 phosphocreatine (disodium salt), pH 7.2 and 290 mOsm. The liquid junction potential correction for this solution was −13 mV. Signals were acquired using pCLAMP 9 recording software and a Multiclamp 700 A amplifier (Molecular Devices, Sunnyvale, CA), sampled at 20 kHz and low-pass filtered at 2 kHz.

RGC dendritic stratification was visualized by including 20 μM Alexa Fluor 594 (Invitrogen, Grand Island, NY) in the intracellular solution. The dendritic morphology of dye-injected RGCs was reconstructed by two-photon imaging with the laser tuned to 780 nm. Images (RGCs and whole retina) were acquired at z intervals of 0.5 μm using a 60× objective (Olympus 60×, 1 NA, LUMPlanFLN). Images were later reconstructed from image stacks with ImageJ.

## Pharmacology

For pharmacology experiments, recordings were performed 10 min after perfusion of ACSF with pharmacological agents at the following concentrations into the recording chamber: 25 μM DL-TBOA, 50 μM D-AP5, 20 μM DNQX, 100 μM MCPG, 100 μM MTEP, 8 μM LY341495, 50 μM atropine, and 100 μM suramin hexasodium salt. All compounds were purchased from Tocris Bioscience, Minneapolis, MN and prepared in dH$_2$O and diluted to the final concentration in ACSF. DL-TBOA was prepared in 0.1% DMSO.

## Focal agonist stimulation

Direct activation of receptors on MCs was accomplished by repeated short applications (100 ms) of L-glutamate (1 mM, Sigma-Aldrich, St Louis, MO), acetylcholine (ACh, 1 mM, Sigma-Aldrich, St Louis, MO), ATP (1 mM, Tocris Bioscience, Minneapolis, MN), or ACSF to the IPL via a pipette that was similar to the recording pipette and a Picospritzer (PV830 Pneumatic PicoPump, World Precisions Instruments, Sarasota, FL) set at 20–30 psi.

## Image analysis

Videos were analyzed offline using SARFIA (freely available on http://www.igorexchange.com/project/SARFIA), a suite of macros running in Igor Pro (Wavemetrics, Portland, OR) (*Dorostkar et al., 2010*) Prior to analysis, images were registered to correct movements in the X and Y directions. Videos were rejected if the plane of focus altered significantly during imaging acquisition. For GLAST*CreER:: GCaMP3* and ShH10-CMV-iGluSnFR recordings, ROIs containing both stalks and lateral processes of MCs were chosen using a filtering algorithm based on a Laplace operator and segmented by applying a threshold, as described in detail in (*Dorostkar et al., 2010*). This algorithm defined most or all of the ROIs that an experienced observer would recognize by eye. For AAV9-2YF-hSynapsin1-iGluSnFR recordings, the entire field of view (256 × 256 pixels) was divided into 16 identical squares and each square was considered one ROI. In both cases, individual ROI responses were then normalized as the relative change in fluorescence (ΔF/F), smoothed by binomial Gaussian filtering, and analyzed to detect activity using custom-made scripts based on a first derivate detection algorithm. A threshold set at ~2 times (when imaging at 0.74 Hz) or at ~4 times (when imaging at 1.7 Hz) the standard deviation of the time derivative trace was used to detect changes in fluorescence within the ROIs. The reliability of this algorithm to detect calcium and glutamate activity was first tested by comparing the results with manual activity detection. For the experiments using focal application of agonists, a threshold set at ~2–3 times the standard deviation of the derivate trace was used to detect activity. In this paper, the fluorescent intensity of ROIs is reported as the average intensity across all pixels within its area. Fluorescent responses are reported as normalized increases as follows:

$$\Delta F/F = (F - F_o)/F_o,$$

where F is the instantaneous fluorescence induced by a spontaneous activity or by a focal application of the agonist and $F_o$ is the baseline fluorescence.

## Statistical analysis

Group measurements are expressed as mean ±SD. We used *t* tests to compare two groups and one-way ANOVA to compare more than two groups. The level of significance was set at p < 0.05.

## Acknowledgements

This work was supported by National Institutes of Health (NIH) grants MH084020 and NS050274 (DB), RO1EY019498, RO1EY013528, and P30EY003176 (JR, RB, GS, MBF). F32EY023160 (GS) National Institutes of Health NRSA Trainee appointment on grant number T32 GM 007232 (GS), R01 EY024958, R01EY022975, and the Foundation Fighting Blindness, USA (JGF, CF). Some images were acquired in Molecular Imaging Center. We thank Dr Ju Chen for providing the $IP_3R2$ knock-out mice.

## Additional information

### Funding

| Funder | Grant reference | Author |
| --- | --- | --- |
| National Institutes of Health (NIH) | RO1EY019498 | Juliana M Rosa, Rémi Bos, Georgeann S Sack, Marla B Feller |
| National Institutes of Health (NIH) | RO1EY013528 | Juliana M Rosa, Rémi Bos, Georgeann S Sack, Marla B Feller |
| National Institutes of Health (NIH) | RO1MH084020 | Dwight E Bergles |
| National Institutes of Health (NIH) | NS050274 | Amit Agarwal, Dwight E Bergles |
| National Institutes of Health (NIH) | R01 EY024958, R01EY022975 | Cécile Fortuny, John G Flannery |
| National Institutes of Health | F32EY023160 | Georgeann S Sack |
| National Institutes of Health | T32 GM 007232 | Georgeann S Sack |
| National Institutes of Health | P30EY003176 | Juliana M Rosa, Rémi Bos, Georgeann S Sack, Marla B Feller |
| Foundation for Preventing Blindness | | Cécile Fortuny, John G Flannery |

The funder had no role in study design, data collection and interpretation, or the decision to submit the work for publication.

### Author contributions

JMR, RB, Conception and design, Acquisition of data, Analysis and interpretation of data, Drafting or revising the article; GSS, Conception and design, Acquisition of data, Drafting or revising the article; CF, JGF, Conception and design, Analysis and interpretation of data, Drafting or revising the article, Contributed unpublished essential data or reagents; AA, Contributed to acquisition of data for Figure 6, Conception and design, Acquisition of data, Drafting or revising the article, Contributed unpublished essential data or reagents; DEB, Provided IP3R2 ko mouse, floxed-GCaAMP3 mouse and data for Figure 6, Conception and design, Acquisition of data, Analysis and interpretation of data, Drafting or revising the article, Contributed unpublished essential data or reagents; MBF, Conception and design, Analysis and interpretation of data, Drafting or revising the article

### Ethics

All animal procedures were approved by the University of California Berkeley and Johns Hopkins University's Animal Care and use Committees and conformed to the NIH Guide for the Care and Use of Laboratory Animals, the Public Health Service Policy, and the SFN Policy on the Use of Animals in Neuroscience Research.

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
