## [Decision Letter]

Thank you for submitting your work entitled “Neuron-glia signaling in developing retina mediated by neurotransmitter spillover” for peer review at *eLife*. Your submission has been favorably evaluated by Gary Westbrook (Senior Editor), a Reviewing Editor, and two reviewers.

The reviewers have discussed the reviews with one another and the Reviewing Editor has drafted this decision to help you prepare a revised submission.

1) In Figure 6, it is shown that P9 retina from IP_3_R2 KO mice still show wave-induced calcium responses, which is interpreted as meaning release of calcium from intracellular stores is not involved in the calcium response at this age. Are the calcium responses present in P7 retina from IP_3_R2 KO mice when the pharmacology suggests that acetylcholine acts through GPCR muscarinic receptors?

2) A series of experiments are used to determine if glutamate acts through AMPA receptors on Muller glia to increase intracellular calcium, rather than an indirect effect of glutamate on neuronal AMPA receptors. However, without doing the genetic experiment of removing AMPA receptors specifically from Muller glia, it is hard to make this conclusion. Please include a statement clarifying this in the text.

Minor points:

1) The authors only briefly discuss their method for imaging only MC processes. Given the possibilities that Cre could be induced in late progenitors at their early experimental time points, and that their dye loading experiments are nonspecific, could they elaborate on how they can be certain that their optical signals originate solely in MCs?

2) Similarly to the first point, could the authors comment on the extent of reporter expression? Are most/all MCs hit, or a subset? Also, in Figure 1 in the GLAST::tdTomato channel, it appears that MC cell bodies are clearly visible in the INL. Why is this not the case for the GLAST*CreER::GCaMP3* (Figure 1), or the ShH10-CMV-iGluSnFR (Figure 4—figure supplement 1)?

3) The bulk of the Discussion is spent discussing possible connections between MCs, Bergmann glial cells, and astrocytes generally. This commentary is intriguing but is perhaps too long for the current submission. It may be more informative to reduce some of the speculation about Bergmann cells in particular, and add a bit more information about our current understanding of the role of Muller cells, and how activity could moderate or guide their development and function.

4) The reference for expression of iGluSnFR in neurons (13) appears to be incorrect.

---

## [Author Response]

*1) In*
Figure 6
*it is shown that P9 retina from IP*_*3*_*R2 KO mice still show wave-induced calcium responses, which is interpreted as meaning release of calcium from intracellular stores is not involved in the calcium response at this age. Are the calcium responses present in P7 retina from IP*_*3*_*R2 KO mice when the pharmacology suggests that acetylcholine acts through GPCR muscarinic receptors?*

We discussed this point with the editor, who, after consulting with the reviewer, decided that an additional experiment would not be required for acceptance. These experiments are non-trivial for the members of the Feller lab since it requires them to travel to Johns Hopkins. As suggested by the editor, we have modified the text in the third paragraph of the Discussion to include this comparison to previous studies that have been done in the CNS:

“Moreover, the mechanism of interaction between neurons and MCs we observed during cholinergic waves is similar to previous findings in visual cortex and hippocampus where cholinergic stimulation leads to astrocyte calcium transients via activation of muscarinic acetylcholine receptors and are abolished in IP_3_R2-KO mice ([Bibr bib8a]; [Bibr bib31b]).”

2) A series of experiments are used to determine if glutamate acts through AMPA receptors on Muller glia to increase intracellular calcium, rather than an indirect effect of glutamate on neuronal AMPA receptors. However, without doing the genetic experiment of removing AMPA receptors specifically from Muller glia, it is hard to make this conclusion. Please include a statement clarifying this in the text.

We have added a sentence qualifying our result in the subsection “MC calcium transients correlated with glutamatergic retinal waves are limited by the extent of glutamate spillover and are mediated by AMPA receptors”:

“Our results strongly suggest that glutamate acts directly on MC AMPARs to increase intracellular calcium. However, it will be necessary to selectively delete AMPARs from MCs to establish their contribution to MC calcium signaling during this period of development.”

*Minor points*:

1) The authors only briefly discuss their method for imaging only MC processes. Given the possibilities that Cre could be induced in late progenitors at their early experimental time points, and that their dye loading experiments are nonspecific, could they elaborate on how they can be certain that their optical signals originate solely in MCs?

We have added the subsection “Calcium-dye loading of MCs” to explain our methods for determining the specificity of loading of organic calcium dyes and expression of genetically encoded reporters.

*2) Similarly to the first point, could the authors comment on the extent of reporter expression? Are most/all MCs hit, or a subset? Also, in*
Figure 1
*in the GLAST::tdTomato channel, it appears that MC cell bodies are clearly visible in the INL. Why is this not the case for the Glast*CreER::GCaMP3 *(*Figure 1*), or the ShH10-CMV-iGluSnFR (*Figure 4—figure supplement 1*)?*

To address the first question, we have added a low magnification image of a transfected retina to Figure 1—figure supplement 1, which shows the effectiveness of expression across the retina. We have also added the following text to the Results:

“There was not a dramatic difference in the density of labeling when we compared genetically labeled versus AM dye loaded MCs, though this was not compared quantitatively.”

To address the second point regarding the visibility of cell bodies, we have added the following text to the Methods:

*“*Note, the labeling in the somas for the GLAST*CreER*::tdTomato […] where signal strength was reduced as a function of depth.”

To address this same issue for ShH10-CMV-iGluSnFR, we have added the following text to Results:

*“*The iGluSnFR signal is proportional to the amount of glutamate reaching the membrane (30); thus, the lack of fluorescence in the INL where MC cell bodies are located indicates that during early development there is little glutamate release outside of the IPL.”

To clarify this point for reviewers, we transfected GFP in MCs using the same AAV variant (ShH10) to illustrate the uniform expression of GFP along the entire MC, including expression in the soma (see Figure 7).

Author response image 1.**DOI:**
http://dx.doi.org/10.7554/eLife.09590.020

*3) The bulk of the Discussion is spent discussing possible connections between MCs, Bergmann glial cells, and astrocytes generally. This commentary is intriguing but is perhaps too long for the current submission. It may be more informative to reduce some of the speculation about Bergmann cells in particular, and add a bit more information about our current understanding of the role of Muller cells, and how activity could moderate or guide their development and function*.

Discussion regarding correlations between MCs and BGs has been shortened. In addition, we have modified the text and opened a discussion point about the possible roles of neuronal activity in the development of Müller glial cells:

“Whether neuronal activity during retinal waves guides the development of MCs is still unclear. […] Our findings in the developing retina differ from studies performed in the adult retina, in which glutamate does not evoke calcium transients in MCs (34; 36), suggesting that glutamaterigic signaling through AMPARs in these cells may be limited to the early stages of synapse formation and refinement.”

*4) The reference for expression of iGluSnFR in neurons (*[13]*) appears to be incorrect*.

This has been corrected.